# Temporal transcriptional response to ethylene gas drives growth hormone cross-regulation in *Arabidopsis*

Katherine Noelani Chang[1], Shan Zhong[2], Matthew T Weirauch[3], Gary Hon[4], Mattia Pelizzola[1†], Hai Li[1], Shao-shan Carol Huang[1,5], Robert J Schmitz[1], Mark A Urich[1], Dwight Kuo[6], Joseph R Nery[7], Hong Qiao[1], Ally Yang[3], Abdullah Jamali[1], Huaming Chen[7], Trey Ideker[8], Bing Ren[4,9], Ziv Bar-Joseph[2,10], Timothy R Hughes[3], Joseph R Ecker[1,5]*

[1]Plant Biology Laboratory, and Genomic Analysis Laboratory, The Salk Institute for Biological Studies, La Jolla, United States; [2]Lane Center for Computational Biology, School of Computer Science, Carnegie Mellon University, Pittsburgh, United States; [3]Department of Molecular Genetics and Banting and Best Department of Medical Research, University of Toronto, Ontario, Canada; [4]Ludwig Institute for Cancer Research, University of California, San Diego, La Jolla, United States; [5]Howard Hughes Medical Institute, The Salk Institute for Biological Studies, La Jolla, United States; [6]Department of Bioengineering, Department of Medicine, The Institute for Genomic Medicine, University of California, San Diego, La Jolla, United States; [7]Genomic Analysis Laboratory, The Salk Institute for Biological Studies, La Jolla, United States; [8]Departments of Medicine and Bioengineering, University of California, San Diego, San Diego, United States; [9]Department of Cellular and Molecular Medicine, University of California, San Diego School of Medicine, San Diego, United States; [10]Machine Learning Department, School of Computer Science, Carnegie Mellon University, Pittsburgh, United States

*For correspondence: ecker@ salk.edu

†Present address: Center for Genomic Science of IIT@SEMM, Istituto Italiano di Tecnologia (IIT), Milan, Italy

Competing interests: The authors declare that no competing interests exist.

**Abstract** The gaseous plant hormone ethylene regulates a multitude of growth and developmental processes. How the numerous growth control pathways are coordinated by the ethylene transcriptional response remains elusive. We characterized the dynamic ethylene transcriptional response by identifying targets of the master regulator of the ethylene signaling pathway, ETHYLENE INSENSITIVE3 (EIN3), using chromatin immunoprecipitation sequencing and transcript sequencing during a timecourse of ethylene treatment. Ethylene-induced transcription occurs in temporal waves regulated by EIN3, suggesting distinct layers of transcriptional control. EIN3 binding was found to modulate a multitude of downstream transcriptional cascades, including a major feedback regulatory circuitry of the ethylene signaling pathway, as well as integrating numerous connections between most of the hormone mediated growth response pathways. These findings provide direct evidence linking each of the major plant growth and development networks in novel ways.

## Introduction

Despite the importance of the plant hormone ethylene, we lack a comprehensive understanding of how its linear signaling pathway mediates many different morphological responses. The dynamic ethylene physiological response, a rapid growth inhibition independent of the master transcriptional regulator

**eLife digest** All multicellular organisms, including plants, produce hormones—chemical messengers that are released in one part of an organism but act in another. The binding of hormones to receptor proteins on the surface of target cells activates signal transduction cascades, leading ultimately to changes in the transcription and translation of genes.

Ethylene is a gaseous plant hormone that acts at trace levels to stimulate or regulate a variety of processes, including the regulation of plant growth, the ripening of fruit and the shedding of leaves. Plants also produce ethylene in response to wounding, pathogen attack or exposure to environmental stresses, such as extreme temperatures or drought. Although the effects of ethylene on plants are well documented, much less is known about how its functions are controlled and coordinated at the molecular level.

Here, Chang et al. reveal how ethylene alters the transcription of DNA into messenger DNA (mRNA) in the plant model organism, *Arabidopsis thaliana*. Ethylene is known to exert some of its effects via a protein called EIN3, which is a transcription factor that acts as the master regulator of the ethylene signaling pathway. To identify the targets of EIN3, Chang et al. exposed plants to ethylene and then used a technique called ChIP-Seq to identify those regions of the DNA that EIN3 binds to. At the same time, they used genome-wide mRNA sequencing to determine which genes showed altered transcription.

Over the course of 24 hr, ethylene induced four distinct waves of transcription, suggesting that discrete layers of transcriptional control are present. EIN3 binding also controlled a multitude of downstream transcriptional cascades, including a major negative feedback loop. Surprisingly, many of the genes that showed altered expression in response to EIN3 binding were also influenced by hormones other than ethylene.

In addition to extending our knowledge of the role of EIN3 in coordinating the effects of ethylene, the work of Chang et al. reveals the extensive connectivity between pathways regulated by distinct hormones in plants. The results may also make it easier to identify key players involved in hormone signaling pathways in other plant species.

ETHYLENE INSENSITIVE3 (EIN3), followed by an EIN3-dependent sustained growth inhibition, calls for a temporal study of ethylene transcriptional regulation (*Binder et al., 2004a*). EIN3 has been shown to be necessary and sufficient for the ethylene response and accumulates upon a duration of exogenous ethylene gas treatment (*Guo and Ecker, 2003*). Although hundreds of ethylene response genes have been identified, because some of the targets of EIN3 are transcription factors (*e.g. ETHYLENE RESPONSE FACTOR1* [*ERF1*]), it is challenging to distinguish immediate early targets from those further downstream. To understand the dynamics of the EIN3-mediated ethylene transcriptional response, we performed a genome-wide study of the ethylene-induced EIN3 protein-DNA interactions using chromatin immunoprecipitation followed by sequencing (ChIP-Seq) and simultaneously determined the repertoire of target genes that are transcriptionally regulated by ethylene (mRNA-Seq). Tracing the transcriptional cascade, we asked if EIN3-mediated genes contribute to a component of the ethylene transcriptional response. For a select number of EIN3 targets that are putative transcriptional regulators, DNA-binding motifs were identified using protein binding microarrays (PBM) and the enrichment for these motifs in the promoters of ethylene response genes was determined.

## Results

We performed ChIP-Seq using a native antibody that recognizes EIN3 (*Guo and Ecker, 2003*) as well as mRNA-Seq in three-day-old dark grown seedlings during a timecourse of ethylene treatment (*Figure 1—figure supplements 1, 2*; *Supplementary file 1A*). By stringent analysis of the temporal ChIP-Seq data (see 'Materials and methods'), we identified 1460 EIN3 binding regions in the *Arabidopsis* genome associated with 1314 genes (*Supplementary file 1B*). We refer to genes associated with EIN3 binding regions as EIN3 candidate targets. In the sequences of EIN3 binding regions, we found significant enrichment of the consensus TEIL motif (Hypergeometric $p < 10^{-87}$) (*Kosugi and Ohashi, 2000*), and de novo motif analysis identified the known EIN3 motif (*Figure 1—figure supplement 3*).

We detected three previously described EIN3 targets using our stringent analysis (*Figure 1—figure supplements 3, 4*) (*Solano et al., 1998*; *Konishi and Yanagisawa, 2008*; *Chen et al., 2009*; *Zhong et al., 2009*; *Boutrot et al., 2010*). One example of a known target of EIN3, *EIN3-BINDING F-BOX PROTEIN 2 (EBF2)*, is shown in *Figure 1A*. EBF2 directs the proteolysis of EIN3 and exhibits ethylene-induced transcription (*Figure 1A*), resulting in feedback regulation of the ethylene signaling pathway. Our study identified additional distal EIN3 binding in the *EBF2* promoter region (*Figure 1A*, *Figure 1—figure supplement 4*).

The majority of studies that exist in the literature have shown that EIN3 acts as an activator, and we observed this activation at the genome-wide level (*Figure 1B*). We found that a majority of EIN3 candidate targets that are regulated by ethylene (referred to as EIN3-R) are induced (85%), Moreover, when compared to the regulation of all genes that respond to ethylene, we observed an over-representation of up-regulation of EIN3 candidate targets (*Figure 1B,C*). Interestingly, many EIN3-R are transcription factors (~14%); EIN3 candidate targets are significantly enriched in gene ontology (GO) terms related to transcription factor regulation, confirming that EIN3 activates a transcriptional cascade (*Figure 1—figure supplement 5*; *Supplementary file 1C*) (*Maere, 2005*).

Numerous studies have reported that transcription factor binding does not necessarily coincide with changes in transcription (*Macquarrie et al., 2011*; *Menet et al., 2012*), especially for master regulators targeting other transcription factors or other factors involved in chromatin state regulation. Only about 30% of the EIN3 binding sites were associated with transcriptional changes, but at least two-thirds were not (*Figure 1D*, *Figure 1—figure supplement 2*). EIN3 candidate targets that are not transcriptionally activated may require cofactors to induce a change in expression for a specific environmental response or developmental program. Quantitatively, the changes in EIN3 binding and steady-state transcription upon ethylene treatment do not correlate because the temporal transcription patterns are very diverse (*Figure 2—figure supplement 1*). However, relatively high levels of EIN3 occupancy in etiolated seedlings treated with ethylene indeed correspond to increases in steady-state levels of transcription (*Figure 2A*). In fact, we were able to differentiate the characteristics of EIN3 candidate targets that exhibited a transcriptional response to ethylene from those that do not (*Figure 2A*). EIN3 candidate targets that exhibit increased occupancy and increased levels of transcription (EIN3-R) are functional targets, enriched in gene families with specific functions, for example BZR, TIFY, and bHLH transcription factor families, which play a role in other hormone pathways ($p < 0.05$) (*Figure 2B*). The highest percentage of hormone-associated genes occurs in EIN3 candidate targets that are ethylene-regulated (EIN3-R) (*Figure 2B*, inset), and it is likely that these EIN3-R targets are direct and/or functional. Other EIN3 candidate targets may play roles in different developmental stages/tissue types, or may be under spatial regulation, requiring specific cofactors.

Projection of the dynamic EIN3 binding (ChIP-Seq) onto the transcriptional ethylene response (mRNA-Seq) using the Dynamic Regulatory Events Miner (DREM) (*Ernst et al., 2007*) revealed that the ethylene response occurs in four waves of transcription significantly regulated by EIN3 (Pathway hypergeometric $p < 10^{-10}$) (*Figure 2C*). These waves display distinct temporal transcription behaviors (Hypergeometric $p < 0.001$), and the reduction of transcriptional noise occurs in successive temporal waves (*Figure 2C*, *Figure 2—figure supplement 2*). Genes that were enriched in specific biological functions within these four transcriptional waves include RNA binding/translation (Wave 1, Wave 3), cell wall maintenance (Wave 2), and response to endogenous stimulus (Wave 4). The second wave is also enriched for genes involved in cell wall maintenance, and the expression of these genes steadily increases following 1 hr of ethylene treatment, consistent with kinetics of EIN3-dependent growth inhibition (*Binder et al., 2004a*; *Vandenbussche et al., 2012*).

The four waves of the ethylene transcriptional response each contain a unique subset of EIN3 candidate targets. The first wave is highly variable, lower in steady-state levels of transcription, and it also contains the lowest percentage of EIN3 candidate targets and hormone-related genes (*Figure 2C*). Previous ethylene growth rate inhibition studies have shown that low amounts of ethylene can result in adaptation and desensitization to subsequent ethylene stimulation (*Binder et al., 2004a*, *2004b*) . This first wave may serve as the immediate ethylene response, activating initial ethylene response genes as well as those that serve to desensitize the plant to subsequent ethylene stimulation, but this has yet to be shown. The next three waves of transcription are successively less variable and contain higher percentages of EIN3 candidate targets and hormone-related genes. The four waves of ethylene-induced transcription account for 50% of the transcriptionally ethylene-regulated EIN3 targets (EIN3-R), and the remaining EIN3 candidate targets are distributed among other patterns of

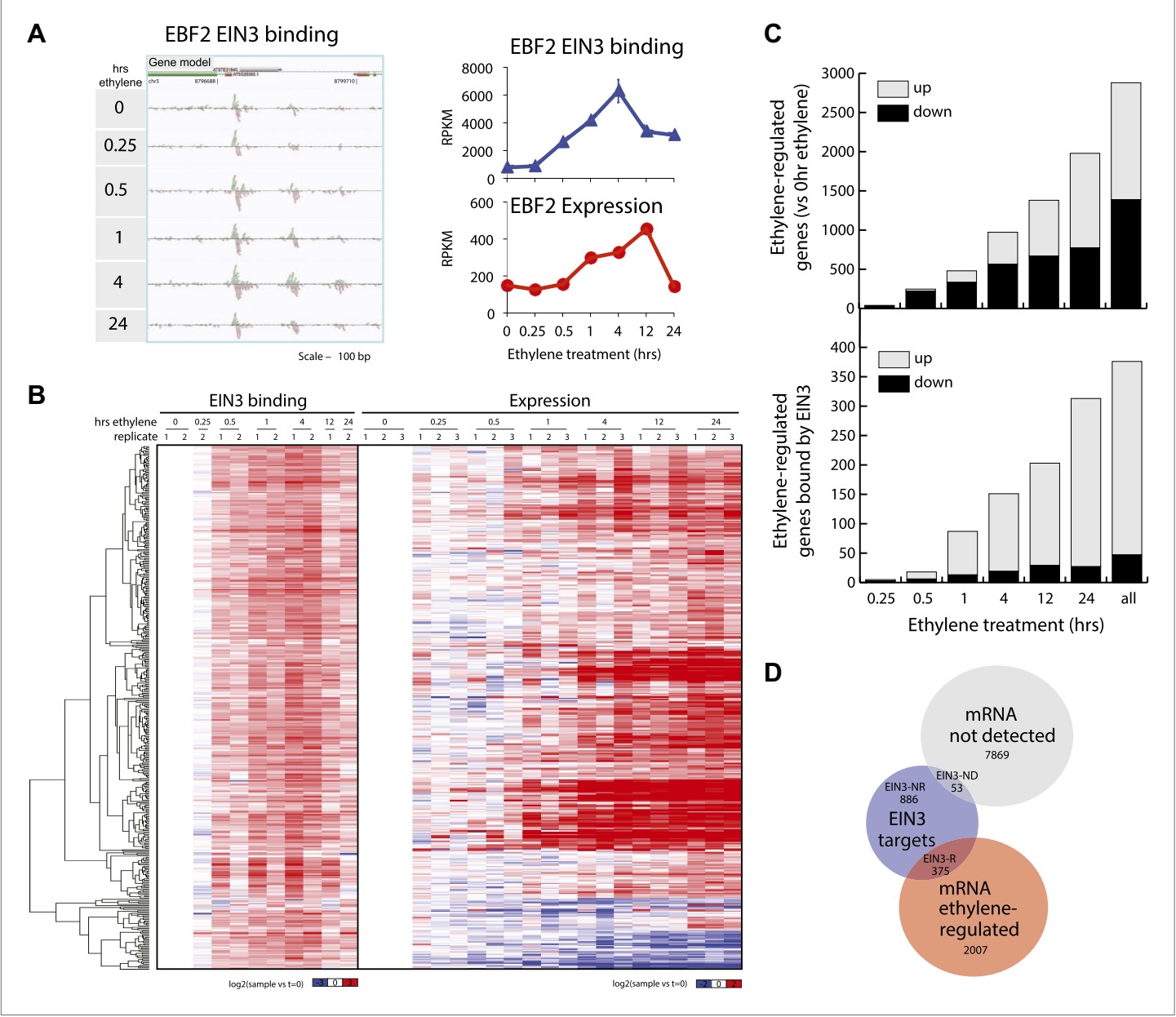

**Figure 1**. Dynamics of ethylene-induced EIN3 binding and transcription supports the role of EIN3 as an activator of the ethylene response. (**A**) Ethylene treatment results in an increase of EIN3 binding in three regions of the EBF2 promoter, corresponding to an increase in steady-state mRNA levels. Binding and transcription levels are indicated by reads per kilobase per million reads in sample (RPKM). Gene model: green (exon), red (UTR), grey (intron/transposon). (**B**) Patterns of EIN3 binding and expression of ethylene-regulated targets are strikingly evident over a timecourse of ethylene gas treatment. EIN3 binding increases with ethylene treatment to a maximum at 4 hr of ethylene treatment for all candidate targets. Each line in the heatmap represents the RPKM value for the representative EIN3 binding site (left panel) and transcript (right panel). (**C**) (Upper panel) Equivalent numbers of genes are up- and down-regulated upon ethylene treatment. (Lower panel) Majority of EIN3 targets differentially expressed upon ethylene treatment are up-regulated. (**D**) A subset of EIN3 targets is transcriptionally regulated by ethylene (EIN3-R).

The following figure supplements are available for figure 1:

**Figure supplement 1**. EIN3 antibody reproducibly enriches DNA in chromatin immunoprecipitation.

**Figure supplement 2**. Ethylene-regulated genes are induced and repressed.

**Figure supplement 3**. Binding of EIN3 to previously known targets.

*Figure 1. Continued on next page*

*Figure 1. Continued*

**Figure supplement 4**. EIN3 ChIP-Seq identified an additional binding in the EBF2 promoter.

**Figure supplement 5**. Functional categories are over-represented for EIN3 targets that are ethylene-regulated (EIN3-R).

transcription that do not contain significant numbers of EIN3 candidate targets in each transcriptional trajectory (Pathway hypergeometric $p<10^{-10}$) (*Figure 2—figure supplement 3*). The expression kinetics and reduction of transcriptional noise we observe in the ethylene-induced waves may be tied to distinct mechanisms of transcriptional control, or they may reflect heterogeneity of the ethylene response in different tissues, which can be resolved using single cell analysis. From the temporal ethylene transcriptional response patterns, it appears that the initial early ethylene transcriptional response is noisy and less focused functionally. During sustained exogenous ethylene application, EIN3 accumulates, and the established ethylene transcriptional response is hormone-focused and less noisy, but feed-forward and feed-back mechanisms mentioned below may serve to establish this functional specificity.

A recurring theme throughout this study is that the key players in the ethylene transcriptional response regulated by EIN3 are involved in plant hormone response pathways, and we anticipate a dense network of interconnections between the coregulated hormone pathways because hormones operate in concert, synergistically/antagonistically regulating growth and development. Although hormone pathway interconnections have been previously described by many groups (*Kaufmann et al., 2009*, *2010*; *Sun et al., 2010*; *Yu et al., 2011*), here we show that these interconnections exist at many regulatory levels and that the targets of EIN3 may regulate genes in these responses. Among the EIN3 candidate targets, we observed the enrichment of hormone-related targets among many different categorical sets (*Figure 2B*, inset). These EIN3 targets include downstream effectors of the ethylene response, key ethylene signaling players, and genes involved in other hormone pathways/responses. Many of the EIN3-modulated downstream effectors are members of the AP2/ERF transcription factor family, and as expected, these transcriptional initiators are up-regulated by ethylene (*Figure 3A*, inset, green font).

Given that EIN3, the master regulator of the ethylene transcriptional response, acts at the culmination of the ethylene signal transduction pathway and is the transcriptional initiator of the ethylene response, one would expect a large number of downstream effectors to coordinate the transcriptional cascade and feedback regulators to maintain the circuitry in a homeostatic state as opposed to a feed-forward runaway response. Analysis of the ethylene-regulated EIN3 targets reveals a number of sites of ethylene signaling modulation of which the majority are negative regulators, supporting the idea that EIN3 is at the end of a signal transduction pathway, and that this regulatory logic dictates a negative feedback loop for homeostatic adaptable systems. More specifically, several negative regulators of the ethylene signaling pathway (*Kendrick and Chang, 2008*) were targets of EIN3 (*Figure 3A*), including three ethylene receptors (ETHYLENE RESPONSE2 [ETR2], ETHYLENE RESPONSE SENSOR1/2 [ERS1/2]), as well as REVERSION-TO-ETHYLENE SENSITIVITY1 (RTE1), CONSTITUTIVE TRIPLE RESPONSE1 (CTR1), and the previously mentioned EBF1/2. The induction of ETR2, ERS1/2 by ethylene was previously reported and has been suggested to restore ethylene receptor activity, resensitizing the plant to ethylene (*Binder et al., 2004b*; *Vandenbussche et al., 2012*). The negative regulation of ethylene signaling by EIN3 through induction of CTR1 and ETR2 is further supported by the literature (*Chen et al., 2007*), suggesting that these proteins exhibit an increase in stabilization upon ethylene treatment (*Gao et al., 2003*).

The EIN3 candidate targets account for more than twice the proportion of hormone genes than in the genome (46%, Hypergeometric $p=10^{-96}$) (*Figure 2B*, inset) (*Alonso et al., 2003a*; *Nemhauser et al., 2006*; *Peng et al., 2008*). Many of the genes were involved in more than one hormone response, highlighting the extensive hormone co-regulation in *Arabidopsis* (*Figure 3B*). Hormone co-regulation is evident in the protein-protein as well as the transcriptional regulator interactions and this network reveals interconnectivity suggestive of robust regulatory co-regulation (*Figure 3C*). Many detailed examples of hormone co-regulation exist in the literature, but often the mechanism(s) of co-regulation is unknown. Previous ChIP-chip or ChIP-Seq studies from plants have also revealed cross-regulation within pathways involved in flowering and in roots (*Yant et al., 2010*; *Iyer-Pascuzzi et al., 2011*;

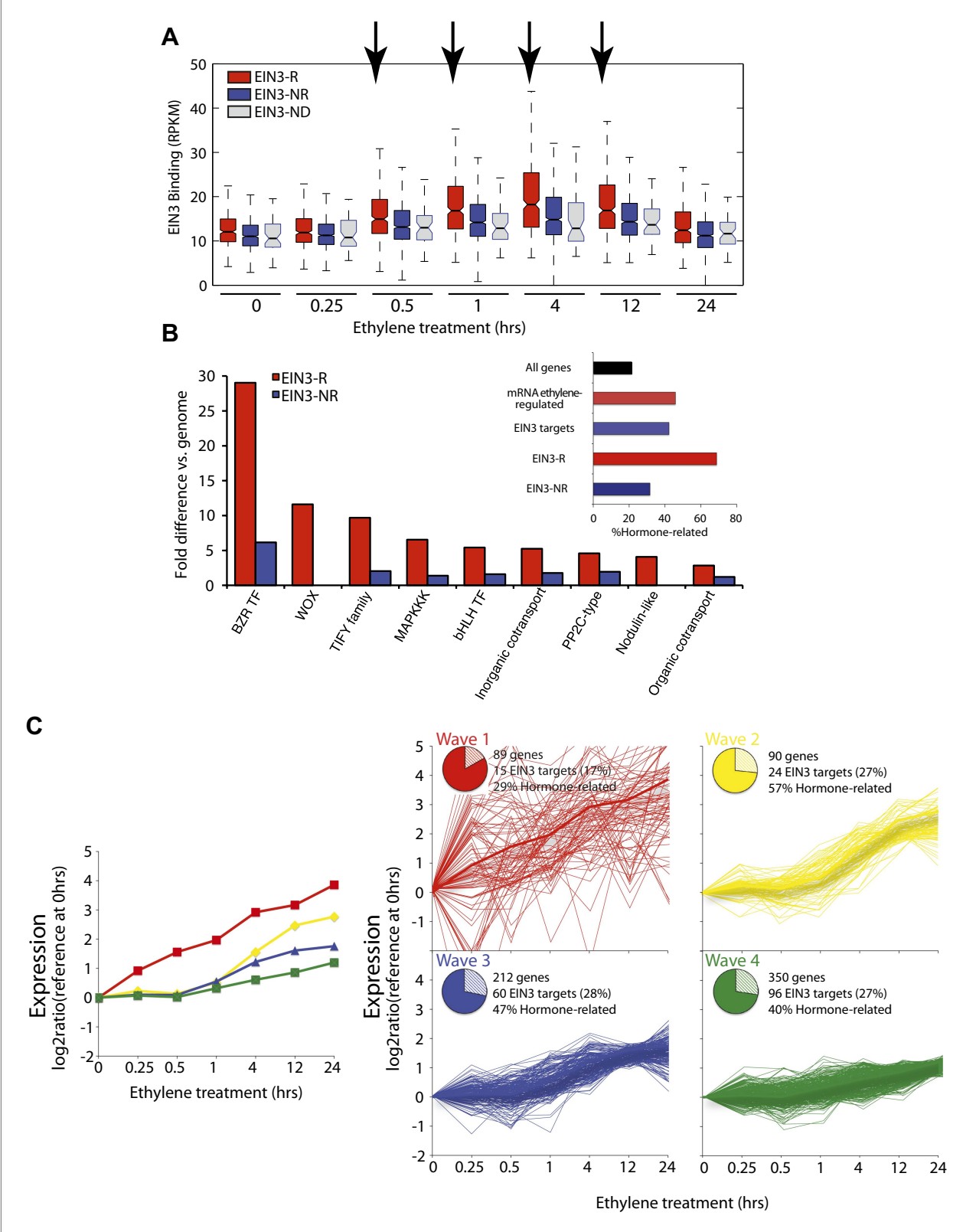

**Figure 2**. The ethylene transcriptional response occurs in four distinct waves of transcriptional induction. (**A**) Ethylene-regulated EIN3 targets (EIN3-R) exhibit increased binding at transcription start sites (TSS) upon ethylene treatment (black arrows) in comparison to those not transcriptionally regulated by ethylene (EIN3-NR and EIN3-ND). Each boxplot represents the distribution of EIN3 ChIP-Seq RPKMs near the TSS. (**B**) Distribution of gene families

*Figure 2. Continued on next page*

*Figure 2. Continued*

among EIN3-R targets reveals over-representation of gene families related to hormone responses function. (Inset) Percentage of hormone-related genes in EIN3 binding and transcription categories. (**C**) DREM paths representing waves of induction of steady-state levels of transcription by ethylene for genes that are regulated by EIN3, implicating different modes of transcriptional regulation in the ethylene response. Right panels contain all genes for each wave.

The following figure supplements are available for figure 2:

**Figure supplement 1**. Quantitative correlation between EIN3 binding and ethylene-regulated expression.

**Figure supplement 2**. Temporal characterization of the ethylene transcriptional response.

**Figure supplement 3**. Ethylene transcription and associated transcription factor regulation kinetics from DREM analysis.

*Winter et al., 2011*; *Immink et al., 2012*) . The findings presented in our study suggest that (1) hormone co-regulation can occur through the binding of EIN3, (2) EIN3 targets hormone pathways at multiple levels, and (3) some of these events are transcriptionally regulated by ethylene (*Figure 3D*).

Ethylene and jasmonate co-regulation occurs at the transcriptional level, sharing a complement of genes responsive to both hormones, for example *RAP2.6L*, *ERF1*. EIN3 also targets four JAZ repressors, two of which are transcriptionally regulated by ethylene (JAZ1, JAZ6). In general, ethylene and jasmonate are known to function synergistically and in the presence of jasmonate, JAZ1 proteins bound to EIN3 are degraded, relieving the EIN3 transcriptional activation (*Zhu et al., 2011*). Here, the presence of an exogenous ethylene stimulus primes cells for a jasmonate response, by loading the promoters of jasmonate/ethylene response genes with EIN3 and JAZ proteins, poising the plant for a jasmonate-ethylene driven transcriptional program, as required for plant pathogen response. Reports of anticipatory binding in other organisms have been forth coming (*Macquarrie et al., 2011*; *Lickwar et al., 2012*).

Ethylene and gibberellin co-regulation through EIN3 occurs at signal reception (GID1B, GID1C) and transcription (PIF3). The regulatory logic of EIN3 binding results in an up-regulation of the gibberellin response; GID receptors target DELLA repressors for degradation, which releases PIF3 from repression, resulting in the activation of the gibberellin transcriptional response. Additional support for feed-forward transcription is provided by over-representation of the PIF3 motif in the promoter sequences of the ethylene transcriptional response genes (*Supplementary file 1E*, *Figure 3—figure supplement 1*). Hormone co-regulation may also occur bidirectionally as a recent study reported negative regulation of ethylene by FUSCA3 (FUS3), known to regulate and be regulated by gibberellin and abscisic acid in embryonic and vegetative timing (*Lumba et al., 2012*). FUS3 negatively regulates genes upstream and downstream of EIN3 (EIN2 and ERF1) in leaf aging (*Lumba et al., 2012*).

Ethylene and auxin co-regulation occurs at both the level of transport and transcriptional response, as EIN3 modulates a regulator of auxin efflux (PID) and its upstream activator (PBP1), and at least seven auxin response proteins (*Supplementary file 1B*). EIN3 also targets the auxin transporter (AUX1) and an auxin signaling gene (IAA29), but these candidate targets are not responsive to ethylene in etiolated seedlings (*Supplementary file 1B*). Ethylene has been reported to stimulate auxin transport through AUX1 away from the root apex, to decrease lateral root primordia (*Lewis et al., 2011*). Therefore, it is likely these binding events have functional outcomes in specific tissue types or developmental programs not addressed in this study.

The establishment of a transcriptional program tailored to result in a specific growth and development process requires multiple levels of transcriptional modulation. EIN3 was previously suggested to initiate a transcriptional cascade because it activates AP2/ERF transcription factors ERF1/EDF1 (*Solano et al., 1998*). To determine additional candidate downstream effectors that may modulate the ethylene transcriptional response cascade, we used in vitro protein binding microarrays to generate DNA-binding motifs for 12 transcription factors that were ethylene-regulated targets of EIN3 (see 'Materials and methods'). We then used the in vitro DNA-binding motifs to scan the promoter sequences of all ethylene transcriptional response genes (*Lam et al., 2011*). EIN3 targets that may regulate a secondary transcriptional ethylene response include AP2/ERFs AT-ERF1, ERF5, and WRKY14/47, PIF3, NAC6, and RAP2.2, and the DNA-binding motifs of the aforementioned transcription factors are

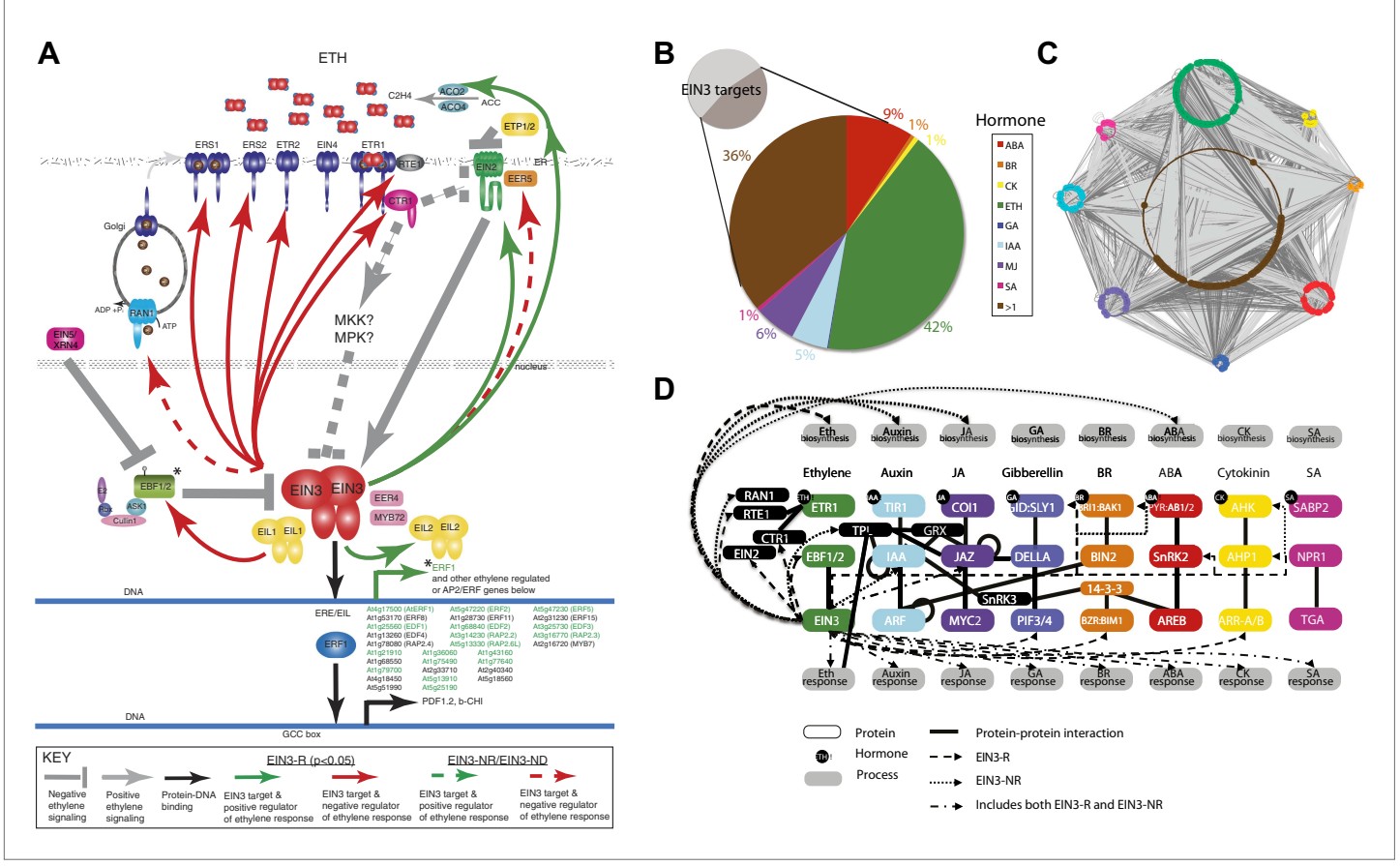

**Figure 3**. Functional classification of EIN3 candidate targets reveals genes involved in hormone responses. (**A**) Feedback (ethylene signaling components, above) of the ethylene response and feedforward (downstream effectors, below). Downstream effectors in green are transcriptionally induced by ethylene. Known EIN3 targets are noted by asterisks; all other EIN3 candidate targets were discovered by this study. (**B** and **C**) EIN3 candidate targets are involved in hormone co-regulation. Node color represents hormone annotation, as indicated in B; large nodes are EIN3 candidate targets. Dark grey edges represent protein-protein interactions (PPI) and light grey edges are protein–DNA interactions (PDI). Hormone annotation legend: abscisic acid (ABA), brassinosteroid (BR), cytokinin (CK), ethylene (ETH), gibberellin (GA), auxin (IAA), methyl jasmonate (MJ), salicylic acid (SA), >1, more than one hormone. (**D**) EIN3-mediated ethylene co-regulation occurs at many different levels. PPIs are from the Arabidopsis Interactome Mapping Consortium, and EIN3 PDIs are from this study.

The following figure supplements are available for figure 3:

**Figure supplement 1**. Motifs of EIN3 targets that are transcriptionally regulated by ethylene were determined in vitro using protein binding microarrays.

over-represented in the promoter regions of genes that are regulated by ethylene (Hypergeometric p<10$^{-5}$) (*Supplementary file 1E*, *Figure 3—figure supplement 1*). Future in vivo analyses of the targets of these transcription factors may help elucidate their contribution to the transcriptional cascade of the ethylene response.

The extensive hormone co-regulation that occurs in waves of transcription leads to certain testable predictions regarding the key regulatory hubs and transcriptional cascades at a genome-wide level. Using a global approach, we are able to determine not only if one gene is a candidate target of EIN3, but whether its homologs are targets as well. Transcription factor targeting of genes that are homologous, with overlapping and unique functions, can add diversity to the outputs of transcriptional programs (*Macquarrie et al., 2011*). One of the most striking and surprising example we found was the direct regulation of the three homologs by EIN3, *HOOKLESS1* (*HLS1*) and *HLS1-LIKE HOMOLOG2* (*HLH2*), and to a lesser extent *HLH1* (*Figure 4A*, *HLH1* in *Figure 4—figure supplement 1*). This led us to experimentally test the functionality of all four members of the *HLS1* gene family in etiolated seedling growth and development. *HLS1* is a well-known signal integrator of ethylene, light, auxin, sugar, and

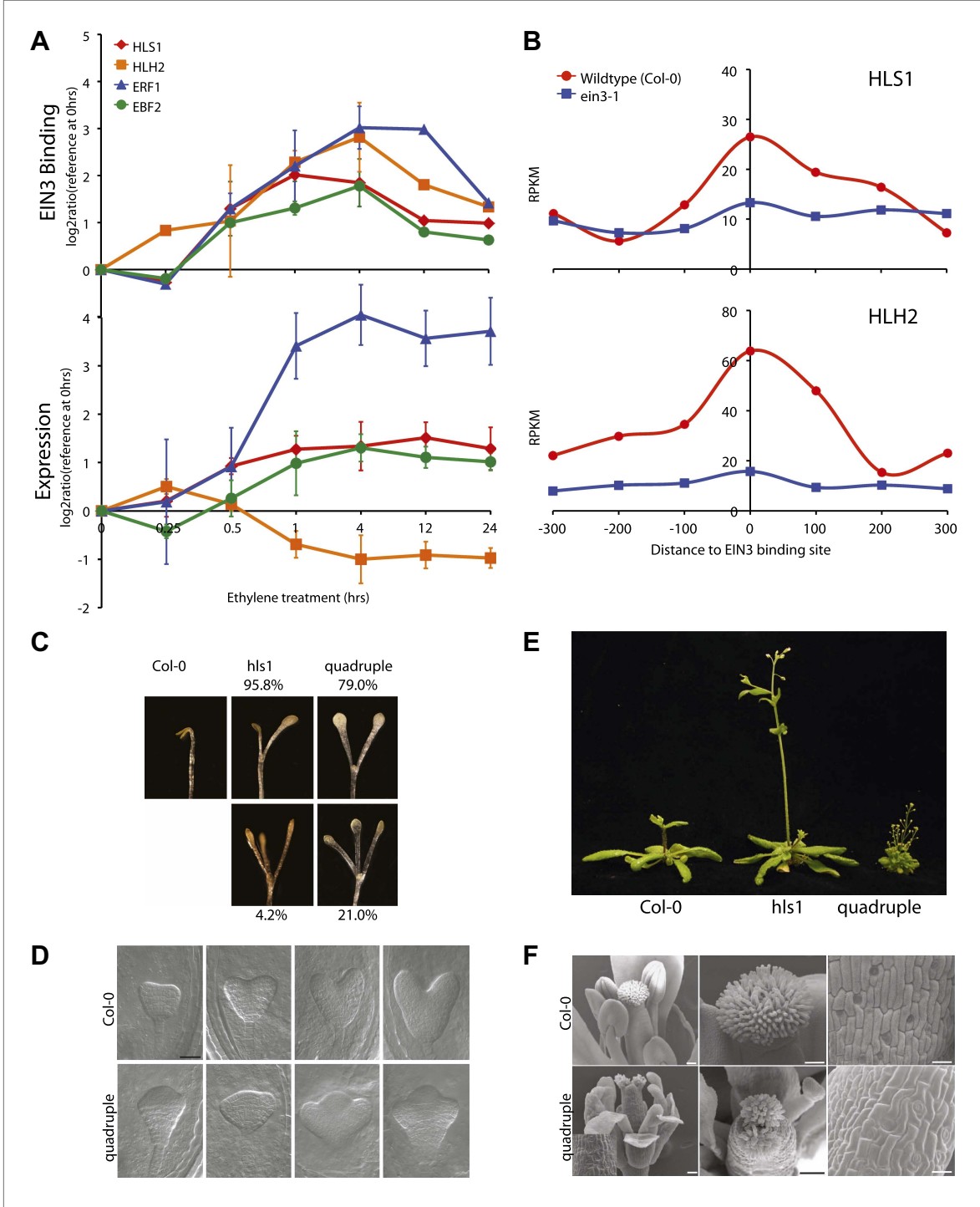

**Figure 4**. EIN3 binding facilitates HLS1 ethylene-auxin hormone co-regulation. (**A**) (Top panel) EIN3 targets *HLS1* and *HLH2*. Temporal EIN3 binding and expression patterns are shown with known EIN3 targets as a control. *HLH1* and *HLH3* are not expressed in etiolated seedlings. (**B**) Binding of EIN3 to *HLS1/HLH2* promoters is dependent on presence of EIN3. (**C**)–(**F**) Mutations in *HLS1* and its homologs reveal severe growth and developmental defects. (**C**) Tri-cotyledon phenotypes in apical hook of quadruple mutants. Images were taken at the same magnification. (**D**) HLS1 gene family has a role in embryo patterning. SEM image scale bar, 50 μm. (**E**) Adult three-week-old plants displayed dwarfed phenotypes similar to *axr1*. (**F**) Quadruple mutants display floral defects similar to *arf3/ettin*. Inset and panels on the right show abnormal guard cell patterning. SEM scale bars, 100 μm.
*Figure 4. Continued on next page*

*Figure 4. Continued*

The following figure supplements are available for figure 4:

**Figure supplement 1**. *HLS1*, *HLH1*, and *HLH2* are targets of EIN3.

**Figure supplement 2**. *HLS1* expression is decreased in *ein3-1*, and *ein3-1/eil1-1* mutants.

**Figure supplement 3**. HLS1-like homologs (HLHs) are similar to HLS1 in protein sequence and domain structure.

**Figure supplement 4**. *Arabidopsis thaliana* EIN3, EIL1, EIL3, and *Physcomitrella patens* EIN3 DNA-binding motifs from protein binding microarray experiments.

brassinolide (*de Grauwe et al., 2005*; *Hou et al., 1993*; *Li et al., 2004*; *Ohto et al., 2006*) and was previously hypothesized to be a target of ERF1 because of the presence of a GCC box motif in the *HLS1* promoter region sequence (*Lehman et al., 1996*). The binding of EIN3 to the promoters of *HLS1*, *HLH2*, and *HLH1* increased upon ethylene treatment (*Figure 4—figure supplement 1*) and is specific to EIN3 (*Figure 4B*). The EIN3 binding sites in these promoters contain known EIN3 motifs (*Figure 4—figure supplement 1*). The functional significance of the *HLS1* EIN3 binding site is supported by a previous study that identified two allelic mutations in the *HLS1* promoter sufficient to yield a 'hookless' phenotype (*Lehman et al., 1996*). Previous studies have also shown that *ein2* is deficient in the accumulation of EIN3 protein (*Guo and Ecker, 2003*) and *HLS1* mRNA, (*Lehman et al., 1996*). We also observed *HLS1* steady-state transcript levels were significantly reduced in the *ein3-1 eil1-1* mutant (*Figure 4—figure supplement 2*).

Ethylene and auxin co-regulate plant growth and development and it is likely that this co-regulation is mediated in part by EIN3 regulation of *HLS1/HLHs*. To understand this hormone co-regulation, we generated quadruple mutants for the *HLS1* gene family and also further characterized their role as regulatory hub signal integrators (*Figure 4—figure supplement 3*). The pleiotropic phenotypes we observed support the role of the *HLS1* gene family in auxin regulated plant growth and development (*Figure 4C–F*). We observed severe defects in the embryonic patterning, etiolated seedlings, adult plant morphology, and floral morphology. The adult quadruple mutants display a dwarf phenotype, similar to the auxin mutant *axr1* (*Leyser et al., 1993*), and floral morphology of the quadruple mutants display two stigmas atop a gynoecium, similar to the *arf3/ettin* mutant floral phenotype (*Sessions and Zambryski, 1995*). Although *HLS1* is known to be involved in the differential growth of the apical hook and is necessary for the accumulation of AUXIN RESPONSE FACTOR2 (ARF2) DNA-binding protein (*Li et al., 2004*; *Ohto et al., 2006*), the biochemical function of these putative N-acetyltransferases remains to be determined. Using a genome-wide approach, we found that not only HLS1, but other gene family members are targets of EIN3 and that the requirement of the HLS proteins for hormone responses extends beyond apical hook development to many other processes from embryo patterning to flowering, linking the regulation of growth and development by ethylene to many new biological processes in novel ways.

## Discussion

To date, few temporal transcription factor binding studies have been undertaken (*Hiroi, 2004*; *Ni et al., 2009*; *Zinzen et al., 2009*). Temporal protein–DNA interactions are often difficult to reconcile with gene expression profiles and the complexity of regulation that occurs transcriptionally is very challenging to characterize and interpret biologically. Here, by jointly analyzing the temporal expression and genome-wide binding data of one key transcription factor in response to hormone stimulus, we were able to reveal several important properties of the hormone responsive transcriptional program and identify new components in the signaling pathway. We found that upon a timecourse of ethylene treatment, EIN3 binding was induced, resulting in various transcriptional patterns and that the ethylene transcriptional response occurred in waves of transcription that were temporally distinct and could be attributed to different biological functions, variable in the amount of noise, and significantly regulated by EIN3. EIN3 modulated genes were over-represented in hormone co-regulation, and the specific targets in the other hormone pathways, as reported in this study, suggest these 'cross-talk' events may involve multiple levels of regulation. Interestingly, feedback regulation of the ethylene response by

EIN3 enabled the identification of the majority of known ethylene signaling pathway components. Moreover, the temporal resolution of steady-state levels of transcription confirmed the role of these genes in the ethylene transcriptional response. Other signaling networks utilize feedback regulation for overall system control/homeostasis, and this type of study may be used to identify novel components in signaling pathways (*Rosenfeld et al., 2002*; *Amit et al., 2007*; *Tsang et al., 2007*; *Avraham and Yarden, 2011*; *Fang et al., 2011*; *Feng et al., 2011*; *Yosef and Regev, 2011*).

The implication that EIN3 regulates the coordination of other hormone pathways is extensive because the transcriptional control by EIN3 is likely conserved in plants. EIN3 orthologs exist in poplar, soybean, rice, maize, moss, and multicellular algae, among many others (www.phytozome.net) and, in fact, we found that the *Physcomitrella patens* (moss) EIN3 protein binds a very similar motif sequence to that of the *Arabidopsis thaliana* EIN3 (*Figure 4—figure supplement 4*). The role of EIN3 in the coordination of the initiation of the ethylene transcriptional cascade, the negative feedback regulation of the ethylene signaling pathway, and the orchestration of other hormone pathways suggests that adaptable system homeostasis in plants requires robust hormone co-regulation.

## Materials and methods

### Plant material

The *Arabidopsis thaliana* ecotype Columbia (Col-0) was the parent strain for these experiments. Genotypes used for this study include wild-type Col-0, and mutants *ein3-1* (*Chao et al., 1997*), *ein3-1/eil1-1* (*Alonso et al., 2003b*), *hls1-1 (hls1)* (*Lehman et al., 1996*), hlh1, hlh2, hlh3 (*Figure 4—figure supplement 3*).

### Growth of *Arabidopsis* seedlings

Three-day-old etiolated seedling tissue was used for these experiments unless otherwise noted. Seeds were sterilized and sown on Murashige and Skoog (cat#LSP03, Caisson) media pH5.7, containing 1% sucrose and 1.8% agar. After stratification for 3 days in the dark at 4°C, exposure to light for 2–4 hr to induce germination, seeds were dark-grown in hydrocarbon free air at 24°C for 3 days. Etiolated seedlings were subsequently treated with ethylene gas at 10 µl l$^{-1}$ for 0, 0.25, 0.5, 1, 4, 12, and 24 hr.

### Chromatin preparation and immunoprecipitation

Etiolated seedlings were collected in the dark, immersed in 1% formaldehyde solution, and cross-linked under vacuum for 15 min. A final concentration of 125 mM glycine was used to quench the formaldehyde for 5 min under vacuum. Cross-linking under vacuum resulted in translucent etiolated seedling tissue. Tissue was liquid nitrogen ground and extraction of chromatin was performed as described in (*Lippman et al., 2005*).

Chromatin immunoprecipitation (ChIP) was performed as described in (*Lippman et al., 2005*) with modifications, including the use of the Bioruptor sonicator (Diagenode, Belgium). Bioruptor settings used were: H, 25 cycles of 0.5 min on, 0.5 min off, with 5 min rests between every 5 cycles. Sonication was performed in a cooling water bath at 4°C. A small amount of chromatin (10 µl) was evaluated for shearing; the size range of chromatin was 150–700 bp, the majority of fragments at 300–400 bp.

Affinity-purified rabbit polyclonal antibodies capable of detecting the C-terminus of EIN3 were used in immunoprecipitation reactions. Details regarding the generation of EIN3 antibodies were previously described (*Guo and Ecker, 2003*). Prior to the experiments in this study, the amount of purified EIN3 antisera per immunoprecipitation reaction was optimized and 8 µl of purified EIN3 antisera was determined to yield the optimal enrichment of the ERF1 promoter, the known target of EIN3 (data not shown). We then substituted Dynabeads Protein A (Invitrogen, cat#100-1D) and Dynabeads M-280 Sheep anti-Rabbit IgG (Invitrogen, cat#112-04D) for the salmon sperm DNA blocked Protein A agarose beads recommended in the protocol (4), as to avoid sequencing of salmon sperm DNA. Immunoprecipitation and washing of Dynabeads were performed using the buffers in (*Lippman et al., 2005*), otherwise Dynabeads were used as per the manufacturer's instructions. Multiple pipetting steps were performed while washing the beads to reduce non-specific binding carryover. Resulting ChIP DNA was purified as in (*Lippman et al., 2005*).

Quantitative PCR revealed that relative ChIP enrichment for the promoter of ERF1 performed with the Dynabeads M-280 Sheep anti-Rabbit IgG was higher in comparison to Dynabeads Protein A

(*Figure 1—figure supplement 1A*). Thus, Dynabeads M-280 Sheep anti-Rabbit IgG was used in all subsequent experiments. Primers for the ERF1 promoter encompassing the EIN3 binding site, are as follows: F-GGGGGCATGTATCTTGAATC, R-TGCTGGATCAACTCAACAAAA. Actin primers were as in Mathieu et al. (*Mathieu et al., 2003*). Enrichment was calculated using the Delta-Delta-Ct method with normalization to the reference Actin; fold change was calculated relative to the control for non-specific binding (EIN3 ChIP performed in *ein3-1* mutant).

ChIP was performed in chromatin derived from wild-type Col-0 three-day-old etiolated seedlings treated with 0, 0.25, 0.5, 1, 4, 12, and 24 hr of ethylene. Two independent biological replicates were used in two replicates experiments for timepoints, 0, 0.5, 1, 4 hr ethylene gas treatment. Single replicates exist for 0.25, 12, 24 hr of ethylene gas treatment.

## Total RNA extraction

Total RNA was extracted from liquid nitrogen ground etiolated seedlings using the Qiagen RNeasy Plant Mini Kit with Qiashredder columns (cat#74,904), with DNaseI (Qiagen, cat#79,254) treatment prior to RNA precipitation in sodium acetate and ethanol. Concentrations of RNA were determined using the ND-1000 spectrometer (Nanodrop, Wilmington, DE). Experiments were performed in three biological replicates for timepoints, 0, 0.25, 0.5, 1, 4, 12, 24 hr ethylene gas treatment.

## ChIP-seq library generation and sequencing

Resulting ChIP DNA from two pooled ChIP reactions above was used to generate a sequencing library as per the Illumina ChIP-Seq manufacturer's instructions. The Illumina Genome Analyzer II (Illumina, San Diego, CA) was used to sequence the single-read ChIP-Seq libraries as per manufacturer's instructions, for 36–43 bps (*Supplementary file 1A*). Raw sequencing data was analyzed using the Genome Analyzer Pipeline v.1.4.0. Reproducibility of the data is shown in *Figure 1—figure supplement 1*. Although the general reproducibility of the data is lower than what was previously reported (*Kaufmann et al., 2009*; *2010*), it is clear that the reproducibility between biological replicates is much higher than that with respect to the control 0 hr ethylene gas treatment timepoint. We did not extend raw reads for calculation of reproducibility but instead determined the reproducibility of RPKM values between replicates.

## PolyA selection and mRNA-Seq library generation

At least 80 μg total RNA was subject to polyA selection using the Poly(A)Purist MAG Kit (Ambion, cat#AM1922). PolyA RNA was subsequently concentrated by ammonium acetate ethanol precipitation and concentrations were determined using the Qubit fluorometer (Invitrogen, Carlsbad, CA) and the Quant-iT RNA Assay Kit (Invitrogen, cat#Q33140). 50–100 ng of polyA RNA was used in a strand-specific library preparation as per the SOLiD Total RNA-Seq Kit protocol (Invitrogen, cat#4445374) and AMPure XP beads (Agencourt, cat#A63881) were used for purification of cDNA and amplified DNA. Samples were barcoded for multiplexing using the SOLiD RNA Barcoding Kit (Invitrogen, Module 1-16 cat#4427046, Module 17-32 cat#4453189, Module 33-48 cat#4453191) as per manufacturer's instructions; final size selection was performed using AMPure XP beads instead of the PAGE purification recommended in the protocol. Size selected libraries were then purified using the MinElute Gel Extraction Kit (Qiagen, cat#28,604). Resulting concentrations of libraries were detecting using the Qubit fluorometer and Quant-iT dsDNA High-Sensitivity Assay Kit (Invitrogen, cat #Q33120). RNA libraries were sequenced for 50 bps on the SOLiD4 platform (Life Technologies, Carlsbad, CA) (*Supplementary file 1A*).

## ChIP-seq data analysis

The Illumina GERALD module was used to align the sequenced reads to the Col-0 reference genome, version TAIR10 (ftp://ftp.arabidopsis.org/). The analysis variable for the ELAND alignment program was set to eland_extended, as read length was greater than 32 bases (e.g., 36–43). Resulting aligned unique single copy reads were used in ChIP-Seq peak analysis (*Supplementary file 1A*).

Saturation analysis of the ChIP libraries was conducted using the spp software (*Kharchenko et al., 2008*) revealed that all samples were at least within 15% of saturation. Peak analysis was performed individually on each timepoint in each biological replicate using the corresponding 0 hr ethylene treated wild-type Col-0 EIN3 ChIP sample as a control. Two additional ethylene treated (4 hr) wild-type EIN3 ChIP biological replicates were included in the analysis, with corresponding mutant *ein3-1* ethylene treated (4 hr) EIN3 ChIP samples as controls. Three software packages: spp (*Kharchenko*

*et al., 2008*), MACS (*Zhang et al., 2008*), PeakSeq (*Rozowsky et al., 2009*) were originally used to identify peaks/regions of binding. Parameters for each software were as follows: MACS (p=0.01), spp (FDR = 0.1), PeakSeq (FDR = 0.1, mingap = 200, minhit = 20, minratio = 3.5). Binding regions were merged when the maximum gap between two peaks was less than 200 bp determined by separate software packages. Subsequent analysis was performed in R. Overlapping peaks in one biological replicate in one timepoint by more than one software package were retained as binding regions. Because of the variation of the number of called peaks in each software and each timepoint, we used a majority vote to call peaks to identify all high stringency EIN3 candidate targets. PeakSeq results differed significantly from spp and MACS (12–76%), therefore only spp and MACS were ultimately used.

Using this method, 1460 EIN3 binding regions were identified (*Supplementary file 1B*). For each EIN3 binding region, the reads per kbp of binding site per million sample reads (RPKM) were calculated. Median normalization of the RPKM values between timecourse biological replicates was performed in R. Resulting RPKMs were log2 transformed with respect to the 0 hr ethylene treatment wild-type Col-0 EIN3 ChIP. Normalization with respect to an input genomic control did not produce distinctively different EIN3 binding pattern profiles (data not shown). EIN3 binding regions were then associated to a gene if located within 5 kbp. The nearest expressed gene (RPKM>1) was assigned if there were more than one gene within 5 kbp. If both genes were not expressed, the nearest gene was selected. Distance was determined from the binding region center to the gene feature using the TAIR10 annotation (ftp://ftp.arabidopsis.org) (*Figure 1—figure supplement 1*).

EIN3 binding profiles of previously determined targets are shown in *Figure 1—figure supplement 3*. Data from biological replicate 1 is shown; biological replicate 2 results were similar. Four of seven previously determined EIN3 targets were identified as EIN3 candidate targets in our dataset. Browser images of data were generated using AnnoJ (*Lister et al., 2008*). ChIP browser images display read tracks normalized per library, the lowest number of reads for all ChIP samples was used as a minimum. This minimum number of reads was randomly selected from all other libraries for display, to effectively visualize enrichment among different samples. The trends in the data were reproducible statistically (*Figure 1—figure supplement 1*), and also evident in the visualization of data (see example of EIN3 binding for both biological replicates in EBF2 promoter depicted in *Figure 1—figure supplement 4*).

Motif identification was performed with the matrix screening software Patser (*Hertz and Stormo, 1999*) and the known EIN3 consensus motif (TEIL) from TRANSFAC previously determined using SELEX (*Kosugi and Ohashi, 2000*). ClustalW2 was used to align motifs (www.ebi.ac.uk/Tools/msa/clustalw2/). Consensus motif representation of the three EIN3 binding sites in the promoter of EBF2 is shown in *Figure 1—figure supplement 4*.

## Gene ontology over-representation of ethylene-regulated EIN3 targets (EIN3-R)

Gene ontology over-representation of selected groups of genes were visualized and determined using the Cytoscape v.2.8.1 (*Shannon et al., 2003*) plugin BiNGO v.2.44 (*Maere, 2005*) (*Supplementary file 1C*). The hypergeometric test was used with Benjamini and Hochberg multiple testing correction (FDR = 0.05). The GOSlim_Plants Ontology was used for *Arabidopsis thaliana* (*Figure 1—figure supplement 5*).

## Motif analysis of EIN3 binding regions

EIN3 binding sites were ranked using the R package timecourse, which has been previously used to analyze microarray timecourse data. We used this R package because no available software to analyze timecourse data for ChIP-Seq data exists. The top 50 EIN3 binding regions were determined and the repeatmasked. De novo motif analysis of these top 50 EIN3 binding regions was performed using SOMBRERO (*Mahony et al., 2005*), and alignment to known *Arabidopsis* motifs (AGRIS, http://arabidopsis.med.ohio-state.edu/) was performed using STAMP (*Mahony and Benos, 2007*) (*Figure 1—figure supplement 3*).

## Protein-binding microarray experiments

Twelve transcription factors that are ethylene-regulated EIN3 targets were analyzed on protein binding microarrays (PBMs). Details of the design and use of universal PBMs has been described elsewhere (*Berger et al., 2006*; *Badis et al., 2009*; *Berger and Bulyk, 2009*). Here, we used two different universal PBM array designs, designated 'ME' and 'HK', after the initials of their designers (*Lam et al., 2011*). Information about individual plasmids is available in *Supplementary file 1D*. We identified the DNA

Binding Domain (DBD) of each TF by searching for Pfam domains (*Finn et al., 2009*) using the HMMER tool (*Eddy, 2009*). DBD sequences along with 50 amino acid residue 'pads' on either side were cloned as SacI–BamHI fragments into pTH5325, a modified T7-driven GST expression vector. Briefly, we used 150 ng of plasmid DNA in a 15 µl in vitro transcription/ translation reaction using a PURExpress In Vitro Protein Synthesis Kit (New England BioLabs) supplemented with RNase inhibitor (Invitrogen) and 50 µM zinc acetate. After a 2 hr incubation at 37°C, 12.5 ml of the mix was added to 137.5 ml of protein-binding solution for a final mix of PBS/2% skim milk/0.2 mg per ml BSA/50 µM zinc acetate/0.1% Tween-20. This mixture was added to an array previously blocked with PBS/2% skim milk and washed once with PBS/0.1% Tween-20 and once with PBS/0.01% Triton-X 100. After a 1 hr incubation at room temperature, the array was washed once with PBS/0.5% Tween-20/50 mM zinc acetate and once with PBS/0.01% Triton-X 100/50 mM zinc acetate. Cy5-labeled anti-GST antibody was added, diluted in PBS/2% skim milk/50 mM zinc acetate. After a 1 hr incubation at room temperature, the array was washed three times with PBS/0.05% Tween-20/50 mM zinc acetate and once with PBS/50 mM zinc acetate. The array was then imaged using an Agilent microarray scanner at 2 mM resolution. Images were scanned at two power settings: 100% photomultiplier tube (PMT) voltage (high), and 10% PMT (low). The two resulting grid images were then manually examined, and the scan with the fewest number of saturated spots was used. Image spot intensities were quantified using ImaGene software (BioDiscovery).

## Motif analysis of EIN3 targets regulating the ethylene transcriptional response

The creation of a position frequency matrix (PFM) from a PBM experiment is non-trivial. For each TF, we therefore evaluated a panel of three algorithms and chose the PFM with the highest performance. For each TF, we ran each algorithm individually on both PBM experiments (HK and ME array designs). The resulting PFMs were then used to score the probe sequences of the opposite array, and these predictions were evaluated based on their Pearson correlation with the actual intensities across all probes. Based on these evaluations, a final PFM was chosen for each TF from the six possible PFMs (three algorithms times two array designs).

We chose three algorithms based on their high performance on an independent PBM dataset (data not shown). Two of the methods, BEEML-PBM (*Zhao and Stormo, 2011*), and FeatureREDUCE (PWM modification of [*Foat et al., 2006*]) are based on biophysical models of TF-DNA interactions. The third algorithm (PWM_align) is an in-house method that aligns all 8mers with E-scores > 0.45 (*Berger et al., 2008*) using ClustalW (*Chenna, 2003*), and trims the resulting alignment by restricting to positions present in at least half of the sequences in the alignment.

The presence of these motifs in the promoter region (−1000bp) of genes that were transcriptionally induced/repressed by ethylene was evaluated to find candidate transcription factors that may be involved in regulating the secondary ethylene transcriptional response. The matrix screening software Patser (*Hertz and Stormo, 1999*) was used to scan the promoter region of all genes that were transcriptionally regulated by ethylene, with the PBM motifs (*Figure 3—figure supplement 1*; *Supplementary file 1E*).

## mRNA-seq analysis

The SOLiD Bioscope v.1.3 software was used to align the reads to the Col-0 reference genome TAIR10 (ftp://ftp.arabidopsis.org/). Two perfect matches per location were allowed. Exonic expression was determined (RPKM) using mRNA-Seq reads mapping in exons in the direction of transcription. Genes were denoted as expressed if they contained RPKM values greater than one for at least one biological replicate in one timepoint. Differentially expressed genes were then called (*t*-test p=0.05, 50% difference from prior timepoint of ethylene gas treatment), and log2 normalized with respect to the 0 hr ethylene gas treatment control (*Figure 1—figure supplement 2*). Overlap of up- and down-regulated genes was ~1%.

## Correlation of EIN3 binding and changes in mRNA steady-state levels

EIN3 ChIP candidate targets were classified as ethylene regulated (EIN3-R), non-ethylene-regulated (EIN3-NR), and transcription not detected in etiolated seedlings (EIN3-ND). The heatmap (*Figure 1—figure supplement 2*) revealed that there is a singular binding pattern but various transcription profiles, as displayed in *Figure 2*. Although the majority of EIN3 candidate targets were up-regulated by ethylene, consistent with the previously determined role of EIN3 as an activator, a subset of EIN3

candidate targets was repressed upon ethylene treatment; one instance of EIN3 as a repressor has been previously reported (*Chen et al., 2009*). The correlation of EIN3 binding and ethylene-regulated transcription was calculated at from 4 hr of ethylene treatment (0 hr ethylene as a control), for all EIN3 and EIN3-R (ethylene-induced) targets. The $R^2$ values were much less than 0.50, suggesting a lack of correlation of EIN3 binding levels and ethylene-regulated steady-state transcription.

The kinetics of transcription was determined for all genes that were transcriptionally regulated by ethylene, and EIN3-R, and reflects the previous growth inhibition study kinetics (*Figure 2—figure supplement 2*). The ethylene transcriptional response was further analyzed in context of the dynamic EIN3 binding data. To reconstruct the dynamic regulatory networks that were activated following ethylene treatment, we used the Dynamic Regulatory Events Miner (DREM) (*Ernst et al., 2007*). DREM integrates time-series gene expression data with static transcription factor (TF)—gene interaction data to reconstruct these dynamic networks. DREM searches for bifurcation events; places in the time series data where the expression of one set of genes diverges from the expression of another set, and annotates these events with the TFs that can explain them. This allows us to assign a time of activation to static TF-gene interactions data. To obtain the static interaction data we extracted 11,355 TF-gene interactions from the AtRegNet AGRIS database (*Yilmaz et al., 2010*). In addition, for this work we have extended DREM so that it can utilize temporal EIN3 binding profiles as well as allowing us to identify functional binding events (those with direct impact on expression). This is done by changing the set of targets for EIN3 so that different binding values are used at each time point. For each EIN3 candidate target gene, the average RPKM values from two input control samples at 0 and 4 hr were used as a cutoff to determine whether it was bound by EIN3 or not at each time point. We ran the modified DREM algorithm using the mRNA-Seq data allowing for 3-way splits. We filtered out genes that did not change at least twofold (up or down) at any time point, and we used the default values for all other parameters.

Four temporally distinct (Hypergeometric p<0.001) EIN3-modulated waves of transcription (Pathway hypergeometric $p<10^{-10}$) were observed. There was a variable amount of noise and percentage of hormone-related genes in each wave of transcription (*Figure 2—figure supplement 2*). The comprehensive DREM analysis results are shown (*Figure 2—figure supplement 3*), including all observed patterns of ethylene transcriptional regulation. The over path significance was used to determine whether these waves were regulated by EIN3. A stringent threshold ($10^{-10}$) was used to identify groups of genes with a significant percentage (>15%) of EIN3 candidate targets.

## Generation of hormone co-regulation network

The most current protein-protein interaction network for *Arabidopsis* (*Arabidopsis Interactome Mapping Consortium, 2011*) containing high throughput yeast two hybrid and literature curated data was used as the foundation for the hormone co-regulation network. The protein-DNA interaction network AtRegNet from AGRIS (http://arabidopsis.med.ohio-state.edu/; 7918 nodes, 10,640 edges) included high throughput data (ChIP-chip and ChIP-Seq) for several transcription factors including AGL15, HY5, GL3, AtbHLH15, WRKY53, GL1, E2F, and SEP3 as well as literature curated data (*Yilmaz et al., 2010*). Transcription factor-DNA binding interactions from six additional studies were added, including TGA2 (*Thibaud-Nissen et al., 2006*), AP1 (*Kaufmann et al., 2010*), BES1 (*Yu et al., 2011*), BZR1 (*Sun et al., 2010*), FLC (*Deng et al., 2011*) in addition to our data. This generated a protein-DNA interaction network of 8531 nodes and 11,953 edges, which was then merged with the protein-protein interaction network. Protein–protein interaction and protein–DNA interaction edges were indicated by dark and light grey lines, respectively.

To identify genes associated with a hormone signal or response (e.g., hormone-related), we used the annotation in the Arabidopsis Hormone Database (*Peng et al., 2008*) (http://ahd.cbi.pku.edu.cn/) in addition to other datasets including relevant ethylene microarray studies in etiolated seedlings (*Alonso et al., 2003a*; *Nemhauser et al., 2006*). Hormone annotation attributes were imported into Cytoscape (*Shannon et al., 2003*) and colored according to hormone. The amount of genes involved in hormone responses in the genome was 21% (5729/27,416), whereas the amount of genes involved in our EIN3 target group was 46% (*Figure 1—figure supplement 2*, inset).

## Identification of loss-of-function mutants for the HLS1 homologs

We identified loss-of-function mutants and performed thorough genetic analyses of *HLS1* and its homologs to characterize the effect, if any, these genes have on the ethylene response. Three HLS1 homologs (HLHs) exist in *Arabidopsis* genome. The protein sequences of the HLHs are homologous

to the full-length protein (*Figure 4—figure supplement 3*). Like *HLS1*, these homologs contain acetyltransferase domains at the N-terminal portion of the protein. Phylogenetic analysis of *HLS1*-like genes with acetyltransferase domain containing proteins from various organisms revealed that the HLS1 family of acetyltransferases form a unique plant-specific class (*Figure 4—figure supplement 3*). We isolated the bona fide loss-of-function mutants in the coding regions of the genes for all the HLH genes using the Salk T-DNA mutant collection (*Figure 4—figure supplement 3*) (*Alonso et al., 2003a*). The single knockout mutants of the HLHs exhibited normal apical hook development and had no obvious developmental defects compared to wild type (data not shown), indicating functional redundancy among HLS1 family members.

## Acknowledgements

We are grateful to the Arabidopsis Interactome Mapping Consortium for the early access to the high throughput yeast two hybrid protein-protein interaction network and to Hongwei Guo for generating the EIN3 antibodies. We thank Zachery Smith and Gerald Pao for critical reading of this manuscript, Mary Galli for helping with the generation of *Figure 3D*, and Nancy Benson for helping review and edit the manuscript. We thank Wilfred de Vega, Atina Cote, Mithunah Krishnamoorthy, and Hong Zheng for assistance with PBM experiments. The data reported in this paper are summarized in *Supplementary file 1* and can be viewed at http://neomorph.salk.edu/dev/pages/EIN3.html. Sequencing data has been submitted to NCBI Sequence Read Archive under accession number SRA063695. JRE acknowledges the Division of Chemical Sciences, Geosciences, and Biosciences, Office of Basic Energy Sciences of the U.S. Department of Energy through Grant DOE FG02-04ER15517 for funding the application of the Chip-seq and RNA-seq timecourse experiments and the analysis of the HLS/HLH quadruple mutant and the National Science Foundation through NSF MCB-1024999 for support of the computational analyses used in these studies.

## Additional information

### Funding

| Funder | Grant reference number | Author |
|---|---|---|
| Department of Energy | DE-FG03-00ER15113, DE-FG02-04ER15517 | Joseph R Ecker |
| National Science Foundation | MCB-0924871 | Joseph R Ecker |
| Canadian Institutes of Health Research | MOP-111007 | Matthew T Weirauch |
| National Science Foundation, Plant Systems Biology IGERT | DGE-0504645 | Katherine Noelani Chang |
| The Gordon and Betty Moore Foundation | Grant GBMF3034 | Joseph R Ecker |
| Gates Millenium Scholarship | | Katherine Noelani Chang |
| National Institutes of Health | 1RO1 GM085022 | Shan Zhong, Ziv Bar-Joseph |
| National Institutes of Health, NRSA | F32- HG004830 | Robert J Schmitz |
| The Howard Hughes Medical Institute | | Joseph R Ecker |
| National Science Foundation | MCB-1024999 | Joseph R Ecker |

The funders had no role in study design, data collection and interpretation, or the decision to submit the work for publication.

### Author contributions

KNC, Conception and design, Acquisition of data, Analysis and interpretation of data, Drafting or revising the article, Contributed unpublished essential data or reagents; SZ, GH, S-sCH, DK, BR, Analysis and interpretation of data, Drafting or revising the article; MTW, Drafting or revising the article, Contributed unpublished essential data or reagents; MP, TI, ZB-J, Conception and design, Analysis

and interpretation of data; HL, RJS, Conception and design, Acquisition of data; MAU, JRN, AY, AJ, Acquisition of data, Drafting or revising the article; HQ, Conception and design, Drafting or revising the article; HC, Conception and design, Drafting or revising the article, Contributed unpublished essential data or reagents; TRH, Conception and design, Contributed unpublished essential data or reagents; JRE, Conception and design, Analysis and interpretation of data, Drafting or revising the article

# Additional files

## Supplementary files

• Supplementary file 1. (**A**) Summary of sequencing reads from EIN3 ChIP-Seq and mRNA-Seq experiments. (**B**) EIN3 candidate targets. (**C**) EIN3 candidate target gene distribution of gene ontology terms. (**D**) Protein binding microarray transcription factor information. (**E**) Motifs of EIN3 targets are over-represented in the promoters of ethylene transcriptional response genes.

## Major datasets

The following dataset was generated:

| Author(s) | Year | Dataset title | Dataset ID and/or URL | Database, license, and accessibility information |
|---|---|---|---|---|
| Chang Katherine N, Zhong Shan, Weirauch Matthew T, Hon Gary, Pelizzola Mattia, Li Hai, Huang Shao-shan Carol, Schmit Robert J, Urich Mark A, Kuo Dwight, Nery Joseph, Qiao Hong, Yang Ally, Jamali Abdullah, Ideker Trey, Ren Bing, Bar-Joseph Ziv, Hughes Timothy R, Ecker Joseph R | 2013 | Temporal transcriptional response to ethylene gas drives growth hormone crossregulation in Arabidopsis | SRA063695; http://www.ncbi.nlm.nih.gov/sra/?term=SRA063695 | Publicly available at the Sequence Read Archive (http://www.ncbi.nlm.nih.gov/sra). |

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
