## [Decision Letter]

Thank you for sending your work entitled “Temporal transcriptional response to ethylene gas drives growth hormone cross-regulation in Arabidopsis” for consideration at *eLife*. Your article has been favorably reviewed by Detlef Weigel (*eLife* Deputy Editor) and an outside peer reviewer. Both discussed their comments and the following reflects their combined assessment of your work.

The manuscript by Chang et al. greatly advances our understanding of how the effects of the plant hormone ethylene are mediated at the molecular level. The authors present a coordinated analysis of transcriptional changes and genome-wide binding of the key transcription factor EIN3 (which accumulates upon ethylene treatment and which is both necessary and sufficient for most ethylene responses) during a time course after addition of ethylene. There are very few reports in the literature tracking genome-wide transcription factor binding after a stimulus, and the integration with gene expression measurements greatly increases the inferences that can be drawn from the genome-wide DNA binding studies. This tour de force will set a new bar for the analysis of transcription factor networks.

Using ChIP-Seq and RNA-Seq, the authors analyzed young Arabidopsis seedlings for EIN3 binding and transcriptional modulation after addition of ethylene. Important findings are that there are several discrete waves of transcriptional responses; that a negative feedback loop represses ethylene receptor expression; and that much of the previously known cross-talk between hormone pathways involves EIN3 binding to genes regulated by several hormones. The interaction with other hormone pathways is nicely demonstrated with the knockout of a small family of closely related EIN3 targets, which reproduces aspects of phenotypes seen in mutants deficient in the response to another hormone, auxin.

The authors also introduce a clever shortcut for understanding the regulatory logic of the later waves of ethylene triggered gene expression. They first nominate candidates for transcription factors that themselves are transcriptionally responsive to EIN3 and that mediate these later responses. The authors then use microarrays to identify sequences that 12 such candidate factors can bind to in vitro. These motifs are then confirmed to be overrepresented in the promoters of the late EIN3 targets.

The only major criticism is that the authors should think a bit harder about their nomenclature. It seems that “targets” refer to any genomic sequence that is bound by EIN3 (or any gene that is within a certain distance from an EIN3 binding site). More commonly, “target” is reserved for bona fide repressed or activated genes. For example, the sentence “many EIN3 targets were not transcriptionally ethylene-regulated (EIN3-NR) (67%), while a subset of the EIN3 targets was transcriptionally ethylene-regulated” should be rephrased as “only about 30% of EIN3 binding sites were associated with transcriptional changes; at least two third were not”.

Minor comments:

1) The first wave of transcriptional regulation does not look very much like a set of coordinated genes. There are many genes that appear to move in the opposite fashion at different time points. This would appear to go beyond the “noise” that the authors use to describe it. Either this should be cleaned up in the sense that the genes that are most coordinated are kept and the others are removed, or the authors need to better explain their rationale for keeping this cluster.

2) Results: over-representation of DNA binding motifs for the 12 candidate factors in EIN3 targets: we suspect it matters which wave is examined; please comment.

3) Results: is the binding of EIN3 to promoters of HLS paralogs associated with overall sequence conservation of the promoters?

4) In addition, it would be nice to know if transcript levels of the other family members decreased in an *ein3-1 eil1-1* double mutant background.

5) Similarly, one would want to know if other auxin related functions like root gravitropism or bud emergence were impaired in the quadruple mutant and if any of its phenotypes could be rescued by exogenous addition of auxin and/or ethylene.

---

## [Author Response]

*The only major criticism is that the authors should think a bit harder about their nomenclature. It seems that “targets” refer to any genomic sequence that is bound by EIN3 (or any gene that is within a certain distance from an EIN3 binding site). More commonly, “target” is reserved for bona fide repressed or activated genes. For example, the sentence “many EIN3 targets were not transcriptionally ethylene-regulated (EIN3-NR) (67%), while a subset of the EIN3 targets was transcriptionally ethylene-regulated” should be rephrased as “only about 30% of EIN3 binding sites were associated with transcriptional changes; at least two third were not”*.

We have modified our nomenclature to reflect the commonly used meaning of “target”. Putative EIN3 targets identified by ChIP-Seq are now referred to as “EIN3 candidate targets”. The subset of EIN3 candidate targets that are ethylene-regulated are referred to as “EIN3 targets”.

*Minor comments*:

*1) The first wave of transcriptional regulation does not look very much like a set of coordinated genes. There are many genes that appear to move in the opposite fashion at different time points. This would appear to go beyond the “noise” that the authors use to describe it. Either this should be cleaned up in the sense that the genes that are most coordinated are kept and the others are removed, or the authors need to better explain their rationale for keeping this cluster*.

Although genes in the first wave have lower expression levels and higher noise, EIN3 is still significantly associated with genes in this wave (p<0.001, permutation test). The gene labels in the expression matrix were permuted 1000 times while the TF-gene interaction table remained as the original. None of the 1000 permutations resulted in EIN3 associated with the first wave bifurcation, reassuring that the enrichment of EIN3 with the first path is strong.

It is likely that if these genes were not part of a coordinated set of genes (p<0.001, hypergeometric test), EIN3 regulation (p<10^-10^, pathway hypergeometric test) would not be associated with this set of genes.

*2) Results: over-representation of DNA binding motifs for the 12 candidate factors in EIN3 targets: we suspect it matters which wave is examined; please comment*.

Three of the twelve candidate factors are in two of the waves of transcription that are regulated by EIN3. The nine others occur in other waves of transcription. Note that the two genes that are repressed by ethylene do not contain over-representation of DNA binding motifs in the promoters of ethylene response genes. It is unknown why the DNA binding motif of At3g24120 is not over-represented in ethylene response genes. Further experimentation (ChIP-Seq with antibodies generated to these transcriptions factors) is required to understand these results.AtIDGene Namep-valueWave of EIN3 enrichmentEthylene Induced (+) or Repressed(-)Average log2 RPKM at 4hrs vs. 0hrs ethyleneAverage log2 EIN3 RPKM at 4hrs vs. 0hrs ethyleneAT1G30650WRKY142.76E-44N/A+0.143.14AT1G09530PIF31.29E-41Wave 4+0.702.01AT5G39610NAC61.07E-34Wave 3+1.662.68AT4G01720WRKY473.16E-18N/A+0.261.71AT4G17500AT-ERF16.58E-12N/A+0.651.54AT5G47230ERF54.55E-11N/A+0.532.42AT3G14230RAP2.23.33E-06N/A+0.792.29AT1G43160RAP2.61.32E-03Wave 3+1.482.68AT3G60530GATA42.86E-03N/A+0.661.64AT1G10480ZFP51.00E+00N/A--0.762.79AT3G241201.00E+00N/A+0.501.84AT3G60390HAT31.00E+00N/A--0.641.61

*3) Results: is the binding of EIN3 to promoters of HLS paralogs associated with overall sequence conservation of the promoters*?

Using ClustalW, we aligned the promoters of HLS1, HLH1, HLH2, HLH3 (3000bp upstream) as well as the EIN3 binding sites in these genes (Figure 5). The binding sites of HLS1 and HLH2 are not associated with the overall sequence conservation of the promoters.

*4) In addition, it would be nice to know if transcript levels of the other family members decreased in an* ein3-1 eil1-1 *double mutant background*.

In general expression of HLH1/2/3 are low (RPKM < 5, and qPCR) in etiolated seedlings (Figure 6); the differences in transcript levels in the *ein3-1 eil1-1* mutant background with the wildtype Col-0 cannot be quantified. We know from earlier studies (Lehman et al, Cell 1996) that HLS1 expression is somewhat restricted and constitutive in many tissues. It is possible the homologs of HLS1 play specific and unique roles in other cell types/tissues.

*5) Similarly, one would want to know if other auxin related functions like root gravitropism or bud emergence were impaired in the quadruple mutant and if any of its phenotypes could be rescued by exogenous addition of auxin and/or ethylene*.

We did not specifically observe and/or quantify root gravitropism phenotypes in the quadruple mutant. We observed an increased number of rosette leaves in the quadruple mutant (n=21), in comparison to wild-type plants (n=12). This phenotype suggests an enhanced not impaired axillary bud emergence, which is expected because auxin inhibits axillary bud emergence in wildtype plants. Concurrent mutations in HLS1 and its homologs resulted in various floral phenotypes. Mutants flowered earlier (2 weeks long day/3 weeks short day) in comparison to the wild type (4 weeks long day/8 weeks short day). Additionally, the quadruple mutant plants lacked apical dominance, producing numerous inflorescences that were sterile. The increased number of rosette leaves and the early flowering phenotype indicate that the quadruple mutants have an accelerated development rate than that of wild type. Interestingly, we found that ARF2 protein accumulates in the quadruple mutant. *arf2* mutants were reported to have increased leaf longevity (Lim et al. 2010, J Exp Botany), and as mentioned previously, it appears the quadruple mutant exhibits the converse phenotype, accelerated development. However, the regulatory logic of ARF2 as a repressor of the auxin response is not simple, if it were, the HLS/HLH quadruple mutant resulting in an increase of ARF2 would therefore result in enhanced repression of the auxin response. However, using Biomaps, genes up-regulated in the quadruple mutant were significantly enriched for “response to auxin” function (unpublished observation). Finally, it is currently not possible to apply exogenous auxin with the expectation of proper delivery of the hormone and complementation of the many phenotypes observed. Regarding ethylene, the quadruple mutant is partially insensitive to ethylene and thus cannot be rescued by the hormone.